# Drospirenone promotes apoptosis in ectopic but inhibits proliferation in eutopic human endometrial stromal cells

Thanyarat Wongwananuruk[1], Somsin Petyim[1,2]*, Pawitra Suwannalert[3], Kittima Tungprasertpol[2], Supatra Klaymook[2]

**1** Gynecologic Endocrinology Unit, Division of Reproductive Medicine, Department of Obstetrics and Gynecology, Faculty of Medicine Siriraj Hospital, Mahidol University, Bangkok, Thailand, **2** Center of Reproductive Genetics and Stem cell Biology, Department of Obstetrics and Gynecology, Faculty of Medicine Siriraj Hospital, Mahidol University, Bangkok, Thailand, **3** Department of Obstetrics and Gynecology, Thammasat University, Bangkok, Thailand

\* somsin.pet@mahidol.ac.th, somsin101@yahoo.com

## Abstract

### Background

Endometriosis is a complex gynecological condition characterized by endometrial tissue growing outside the uterus. In many *in vitro* studies, almost all progestins have indicated the anti-proliferation and apoptosis of endometriotic stromal cells. Drospirenone, a synthetic progestin structurally distinct from traditional progestins, still lacks sufficient data regarding its effects on endometriosis, particularly in terms of antiproliferative and pro-apoptotic activity. This study investigates the antiproliferative effects of drospirenone on eutopic (EU-ESCs) and ectopic human endometrial stromal cells (EC-ESCs), and compare its impact on apoptotic effects in both cell types.

### Methods and Findings

In the study, paired EU-ESCs and EC-ESCs were obtained from patients diagnosed with endometriosis (n = 12). EU-ESCs and EC-ESCs were treated with and without drospirenone. Antiproliferative markers and apoptotic markers were evaluated and compared between the two groups. Interestingly, drospirenone at a concentration of 1 μM significantly affected cell viability in both EU-ESCs and EC-ESCs. In EU-ESCs, Ki-67 expression was significantly reduced compared to controls (0.17 vs. 1; p = 0.003), while in EC-ESCs, the reduction was not statistically significant. Caspase-3 expression was significantly increased in both EU-ESCs (1.13 vs. 1) and EC-ESCs (1.57 vs. 1) (p = 0.02 and p = 0.05, respectively). Additionally, *BCL2* expression decreased in both cell types following treatment. *BAX* expression increased in both EU-ESCs and EC-ESCs. Expression levels of *PTEN* and *P53* also increased in both cell types, with statistical significance observed only in EC-ESCs (p = 0.03

**Data availability statement:** All relevant data are within the paper and its Supporting Information file.

**Funding:** This study was supported by the Research Development Grants, Faculty of Medicine Siriraj Hospital, Mahidol University (no. R016631035). The funders had no role in study design, data collection and analysis, decision to publish, or preparation of the manuscript.

**Competing interests:** The authors have declared that no competing interests exist.

**Abbreviation:** EU-ESCs, Eutopic human endometrial stromal cells, EC-ESCs, Ectopic human endometrial stromal cells, DRSP, Drospirenone

and $p = 0.04$, respectively). *BAK* expression decreased in EU-ESCs but increased in EC-ESCs compared to controls.

## Conclusions

Drospirenone exhibits an antiproliferative effect on EU-ESCs and induces a more pronounced apoptotic response in EC-ESCs.

## Introduction

Endometriosis is a complex gynecological condition characterized by endometrial tissue growing outside the uterus, such as ovaries, fallopian tubes, uterus, and peritoneum. It is barely found in extra-pelvic areas, such as the pleura, and diaphragm. This condition affects about 10% of reproductive women and up to 50% of women diagnosed with infertility. The peak incidence is in women aged between 25–45 years. Frequently, the main presenting symptoms are progressive dysmenorrhea, dyspareunia, and chronic pelvic pain [1,2].

In women with endometriosis, there are many hormonal modifications and localized steroidogenesis in both eutopic and ectopic endometrium. Those alterations are due to an increase in aromatase enzyme activity, estrogen receptor (ER)-β isoform but a decrease in 17β-Hydroxysteroid dehydrogenases (HSD) and the progesterone receptor (PR)-β resulting in micro-environment of hyperestrogenism and progesterone resistance. Moreover, various inflammatory cytokines such as interleukins (IL-1 β, IL-6, IL-8), tumor necrosis factor (TNF)-α, and impaired function of macrophage and natural-killer (NK) cells are found to be factors promoting survival and invasion [1–4].

Medical therapy remains the mainstay of endometriosis management. According to the 2022 ESHRE endometriosis guideline [5], combined oral contraceptives and progestin-only therapies are recommended as first-line treatment, acting primarily through the induction of anovulation and endometrial pseudo-decidualization. In addition to contraceptive effects, these agents effectively reduce endometriosis-associated pain. A wide range of progestins is incorporated into oral contraceptive formulations, allowing optimization of therapeutic efficacy while minimizing adverse effects. Drospirenone is a fourth-generation progestin structurally derived from spironolactone, distinguishing it from testosterone- and progesterone-derived progestins. This unique molecular backbone confers combined progestogenic, anti-androgenic, and anti-mineralocorticoid properties, which may differentially affect intracellular signaling pathways involved in cell survival and apoptosis. Drospirenone has recently been approved as a progestin-only pill containing 4 mg of non-micronized drospirenone administered in a 24/4 regimen [6,7].

Progestins exert diverse biological effects that extend beyond progesterone receptor activation, including the modulation of cell proliferation, differentiation, and apoptosis. These effects vary substantially among synthetic progestins due to differences in chemical structure, receptor affinity, and off-target activity. Although several

progestins have been shown to influence apoptotic pathways, the available evidence is largely derived from structurally related compounds, limiting extrapolation across different progestin classes. In many *in vitro* studies, most progestins have been reported to induce apoptosis and exert antiproliferative effects in endometriotic stromal cells [8–10]. Norethisterone, for example, can inhibit ectopic endometrial stromal cell proliferation and induce cell apoptosis [11]. Dienogest, another common potent oral progestin used, can inhibit aromatase and cyclooxygenase-2 expression and prostaglandin E2 production in ectopic human endometrial stromal cells [8,9,12–14]. In drospirenone, drospirenone can reduce inflammatory cytokines, vascular endothelial growth factor (VEGF), and nerve growth factor (NGF) expression in human endometriotic stromal cells mediated via progesterone receptor [15]. Drospirenone can induce decidualization of eutopic endometrium and reduce DNA synthesis in ectopic endometrium which moderated via PR [16].

Despite the widespread clinical use of drospirenone in oral combined hormonal contraception and the management of endometriosis, its direct cellular effects on apoptotic regulation and antiproliferative activity in endometriosis remain incompletely characterized. Addressing this knowledge gap is essential for understanding the biological implications of drospirenone, particularly in hormonally responsive tissues. Therefore, this study aimed to evaluate the antiproliferative effects of drospirenone on eutopic endometrial stromal cells (EU-ESCs) and ectopic endometriotic stromal cells (EC-ESCs), and to compare its effects on apoptosis in these two cell populations.

## Materials and methods

### Human eutopic and ectopic endometrial tissue samples

Samples of eutopic and ectopic endometrium were obtained from women who attended the Department of Obstetrics and Gynecology, Faculty of Medicine, Siriraj Hospital. The inclusion criteria were women aged 18–45 years who were diagnosed with endometriotic cysts and scheduled to undergo cystectomy or adnexectomy, with or without hysterectomy, via laparoscopic or laparotomy approaches. The exclusion criteria included patients who had received hormonal medications, nonsteroidal anti-inflammatory drugs (NSAIDs), or herbal supplements within three months prior to surgery. This study was approved by the Institutional Review Board of the Human Research Protection Unit, Faculty of Medicine, Siriraj Hospital, Mahidol University (SI 150/2023) prior to enrollment. Eligible patients received comprehensive counseling and provided written informed consent before participation. The study was conducted between March 2023 and January 2024.

In all cases, the operation was performed within the first week after menstruation. At the time of operation, endometrial sampling by endocell® or medgyn endosampler® was performed to obtain eutopic endometrium in the proliferative phase. During laparoscopic ovarian cyst enucleation, the endometriotic cyst wall was completely peeled off from the normal ovarian tissue. A portion of the cyst wall (approximately 2 cm²) was collected for experimental use, while the remaining tissue was sent for pathological examination, which confirmed the diagnosis of endometrioma. The specimens whose pathological report could not be concluded with endometriosis were withdrawn from the experiments. All experiments were conducted in triplicate.

### Isolation and culture of eutopic human endometrial stromal cells (EU-ESCs) and ectopic human endometrial stromal cells (EC-ESCs)

For cell isolation and culture of EU-ESCs, endometrial tissues were irrigated in normal saline to remove the blood and then washed with Dulbecco's phosphate-buffered saline (DPBS; Gibco, Invitrogen, CA) containing 2% antibiotic and antimycotic solution (2% penicillin-streptomycin (P-S; Gibco) and 0.5 µg/mL amphotericin B; Gibco). After that, tissues were washed with Dulbecco's Modified Eagle Medium (DMEM; Gibco) combined with 2% antibiotic and antimycotic solution. The tissues were dissected into small pieces and plated to culture dishes. The cultured plates were incubated in DMEM (Gibco), 10% fetal bovine serum (FBS; Gibco), 1% P-S (Gibco) and 0.25 µg/mL amphotericin B (Gibco) at 37ºC in a humidified 5% $CO_2$ for three days and then the media was changed. After cells outgrowth, the tissue was discarded and

medium was change to 10% DMEM containing DMEM (Gibco), 10% FBS (Gibco) and 1% P-S (Gibco). EU-ESCs were expanded to 70–80% confluency with medium changed every 3–4 days.

For EC-ESCs, the tissues of ectopic endometrium from endometriotic cyst wall were irrigated with normal saline and minced into small pieces. They were washed with DPBS and 2% antibiotic and antimycotic solution then washed with DMEM combined with 2% antibiotic and antimycotic solution. The minced tissues were plated into culture dishes and maintained in the same medium and environment as the eutopic endometrium. EC-ESCs were cultured under 10%DMEM with medium changed every 3–4 days until reached their 70–80% confluency. EU-ESCs and EC-ESCs were scaled up by subculture passaged to generate a sufficient amount for the study. To confirm the phenotypic identity of EU-ESCs and EC-ESCs, flow cytometry was performed using mesenchymal stem cell (MSC) markers (CD29, CD44, CD73, CD90, and CD105).

## MTT viability assay of EU-ESCs and EC-ESCs

MTT viability assay was performed to examine the dose-response antiproliferative effect of cells in both EU-ESCs and EC-ESCs. The principle of the MTT viability assay based on the cleavage of the yellow tetrazolium salt MTT (3-(4,5-dimethylthiazol-2-yl)-2,5-diphenyl-2H-tetrazolium bromide) to form a violet formazan. A decrease in the number of living cells results in a decrease in the metabolic activity of the experimented culture. This decrease is directly correlated with the amount of violet formazan formed, as measured by a microplate reader at a wavelength at which MTT-derived formazan absorbs the most (around 570 nm) [17].

EU-ESCs and EC-ESCs were seeded at $1 \times 10^4$ cells per well in a triplicate manner and cultured in 10%DMEM for 24 hours. After the cells reached 70−80% confluence, the medium was changed to DMEM (Gibco) with 2% charcoal-stripped fetal-bovine serum (CS-FBS; Sigma-Aldrich) and 1% P-S (Gibco). After 24 hours of incubation, the initial media was changed into the experiment medium consistent of DMEM (Gibco), 2% CS-FBS (Sigma-Aldrich, MO), 1% P-S (Gibco) supplemented with drospirenone (Sigma-Aldrich) at concentration of 0.1, 1, or 10 µM and then cultured for 24 hours. Cells cultured under 10%DMEM were served as control. The medium was discarded and 10% MTT (AppliChem GmbH, Darmstadt, Germany) solution was added and incubated for 4 hours at 37°C, 5% $CO_2$ in a dark environment. MTT solution was then discarded and DMSO (Sigma-Aldrich) was added. Finally, the experiment cultures were transferred to a microplate reader (Synergy H1, Biotek) at 570 nm wavelength. The concentration of drospirenone that caused about a 50% reduction in absorbance could referred to as an effective concentration to use further for Ki-67 and caspase-3 expression analysis.

## Ki-67 expression analysis by flow cytometry

Ki-67 is a protein encoded by the MKI67 gene that is expressed in almost all of the proliferating cells in both the mitotic phase and interphase, except in G0 [18]. EU-ESCs and EC-ESCs were seeded at $1−1.5 \times 10^4$ cells/cm$^2$ and cultured in 10%DMEM for 24 hours. After the cells reached 70−80% confluence, the media was changed to DMEM (Gibco) with 2% CS-FBS (Sigma-Aldrich) and 1% P-S Gibco) and incubated for 24 hours. After that, the initial media was changed into experiment media with specific concentrations of drospirenone and cultured for 24 hours. Cells cultured under 10%DMEM were served as control medium. Trypsinization was then performed to resuspend the cells preparing for fixation and permeabilization by add 3 ml of cold 70% ethanol and incubate at −20°C for 1 hour. Washing with BioLegend's cell staining buffer (Biolegend, CA). After removal of the excess ethanol, the PE-conjugated Ki-67 and Isotype control antibodies (Biolegend) were added and incubated at room temperature for 30 minutes in dark. Finally, cells were resuspended with 0.5 ml cell staining buffer and analysis of the staining cells by using flow cytometry (FACSCalibur, BD Bioscience).

## Caspase-3 expression analysis by flow cytometry

Caspase is a protein encoded by the CASP gene which is involved in the execution phase of programmed cell death (apoptosis). Among them, caspase-3 is a frequently activated dead protease [19]. The EU-ESCs and EC-ESCs were seeded at $1−1.5 \times 10^4$ cells/cm$^2$ and cultured in 10%DMEM for 24 hours. After the cells reached their 70−80% confluence, the media

was changed to DMEM (Gibco) with 2% charcoal-stripped FBS (Sigma-Aldrich) and 1% antibiotics incubated for 24 hours. After that, the initial media was changed into experiment media with specific concentrations of drospirenone and cultured for 24 hours. Cells cultured under 10%DMEM were served as control medium. Trypsinization was then performed to resuspend the cells preparing for fixation by 4% formaldehyde with 1 μL to 10,000 cells. After 15 minutes of fixation, cells were permeabilized at least 10 minutes by adding ice-cold 100% methanol slowly to pre-chilled cells, while gently vortexing, to a final concentration of 90% methanol. After removal of the excess methanol, the PE-conjugated caspase-3 and Isotype control antibodies (Cell Signaling Technology, MA) were added and incubated at room temperature for 1 h in dark. Finally, cells were resuspended with PBS and analysis of the staining cells by using flow cytometry (FACSCalibur, BD Bioscience).

## Quantitative RT-PCR

The EU-ESCs and EC-ESCs were seeded at $1-1.5 \times 10^4$ cells/cm$^2$ and cultured in 10% DMEM for 24 hours. After the cells reached their 70–80% confluence, the media was changed to DMEM (Gibco) with 2% charcoal-stripped FBS (Sigma-Aldrich) and 1% antibiotics incubated for 24 hours. After that, the initial media was changed into experiment media with specific concentrations of drospirenone and cultured for 24 hours. Cells cultured under 10%DMEM were served as control medium. The total RNA was extracted using TRIzol reagent (Thermo Fisher Scientific, Waltham, MA). cDNA was synthesized using iScript Reverse Transcription Supermix (Bio-Rad Laboratories, Hercules, CA) and amplified using selective primers (Table 1). Quantitative RT-PCR was performed on a LightCycler 480 using SYBR Green I master mix (Roche Diagnostics GmbH, Mannheim, Germany). The data were analyzed using the $2^{-\Delta\Delta CT}$ method relative to the quantity of *beta-actin*.

## Statistical analysis

The continuous data were expressed in mean ± standard error of the mean (SEM). Normality of each dataset was assessed using the Kolmogorov–Smirnov test. *t*-tests were used to compare outcomes between groups. The differences among multiple groups were analyzed by using ANOVA and Bonferroni's post-hoc test. A P-value less than 0.05 was considered to have statistical significance. Data analysis was conducted using SPSS version 23 (SPSS Inc., Chicago, USA).

## Results

### Human eutopic and ectopic endometrial tissue samples and Baseline characteristics

Paired eutopic and ectopic endometrium were obtained from twelve patients diagnosed with endometriotic cyst and underwent the operation. Four of them were withdrawn due to insufficient specimens. Most of them were nulliparous women

**Table 1. Primer sequences and annealing temperatures.**

| Gene | Primer sequences (5' – 3') | Annealing temp. (°C) |
|---|---|---|
| *β-actin* | F: 5'-ATGTGGCCGAGGACTTTGATT-3'<br>R: 5'-AGTGGGGTGGCTTTTAGGATG-3' | 60 |
| *BAK* | F: 5'- CAAGATTGCCACCAGCCTGTTTGA −3'<br>R: 5'- ATGCAGTGATGCAGCATGAAGTCG −3' | 57 |
| *BAX* | F: 5'- CCCGAGAGGTCTTTTTCCGAG −3'<br>R: 5'- CCAGCCCATGATGGTTCTGAT-3' | 57 |
| *BCL2* | F: 5'- GGGTATGAAGGACCTGTATTGG −3'<br>R: 5'- CATGCTGATGTCTCTGGAATCT −3' | 58 |
| *P53* | F: 5'- GAGGTTGGCTCTGACTGTACC −3'<br>R: 5'- TCCGTCCCAGTAGATTACCAC −3' | 56 |
| *PTEN* | F: 5'- TTTGAAGACCATAACCCACCAC −3'<br>R: 5'- ATTACACCAGTTCGTCCCTTTC −3' | 56 |

(87.5%) with a mean age (SD) of 35.12 (7.61) years. Mean BMI (SD) was 21.76 (5.13) kg/m$^2$. The most common presenting symptom was progressive dysmenorrhea accounting for 75% of patients and the remaining 25% was non-cyclic pelvic pain. Endometriotic cyst mean diameter (SD) was 6.68 (2.05) cm. Most of the patients had underwent laparoscopic ovarian cystectomy or oophorectomy (87.5%).

**Effect of drospirenone on EU-ESCs and EC-ESCs viability: Determine drospirenone concentration by MTT viability assay**

EU-ESCs and EC-ESCs were successfully isolated (Fig 1) and continued passage until reaching 70–80% confluence. The cells' morphology is spindle-like in shape (Fig 2). Flow cytometry study using mesenchymal stem cell (MSC) markers (CD29, CD44, CD73, CD90, and CD105) demonstrating positive expression of typical MSC markers (Fig 3). They were then treated with various concentrations of drospirenone. Drospirenone reduced cell viability in all concentrations but significantly reduced in groups treated with 1 and 10 µM of drospirenone in EU-ESCs compared with the control group ($p = 0.03$ and $p = 0.01$, respectively) (Fig 4A). Some differences in cell response in EC-ESCs, the cell viability was reduced by all concentrations with statistical significance compared with the control group ($p = 0.004$ for 0.1 µM vs control, $p = 0.001$ for 1 µM vs control and $p = 0.03$ for 10 µM vs control, respectively) (Fig 4B).

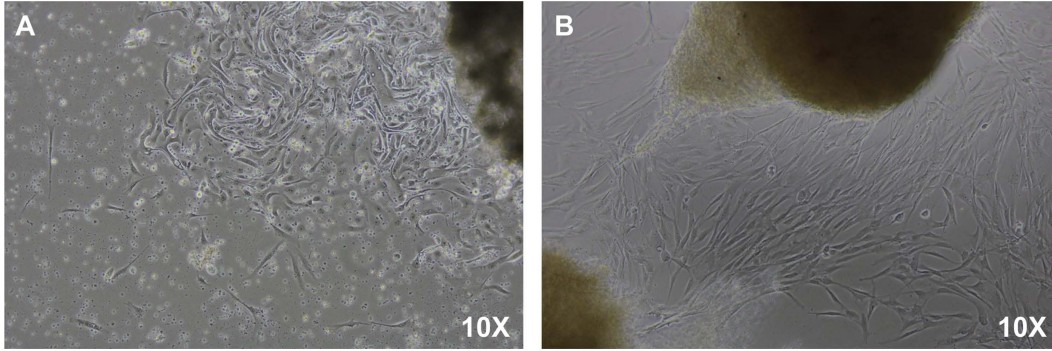

**Fig 1. Cell outgrowth from tissue at day 5 of primary culture. (A)** Eutopic human endometrial stromal cells (EU-ESCs) outgrowth from endometrial tissue under culture medium. **(B)** Ectopic human endometrial stromal cells (EC-ESCs) outgrowth from tissues of ectopic endometrium from endometriotic cyst wall under culture medium.

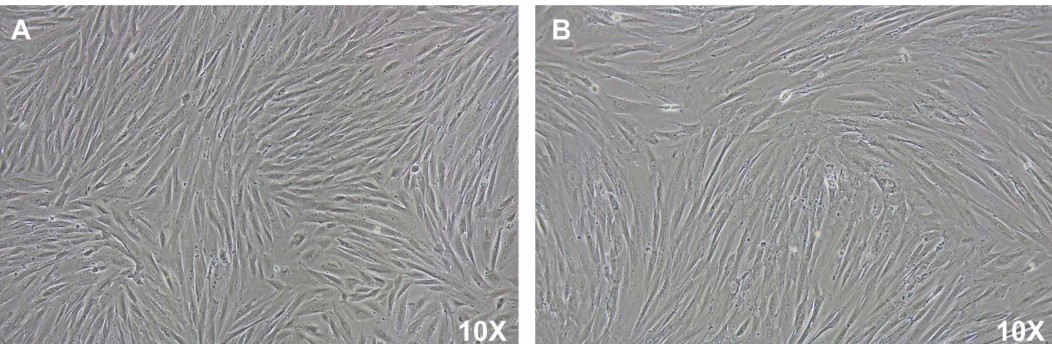

**Fig 2. Human endometrial stromal cells from the patients with endometriosis can be isolated and expanded *in vitro*. (A)** EU-ESCs and **(B)** EC-ESCs at passage 4 were observed under 10X magnification.

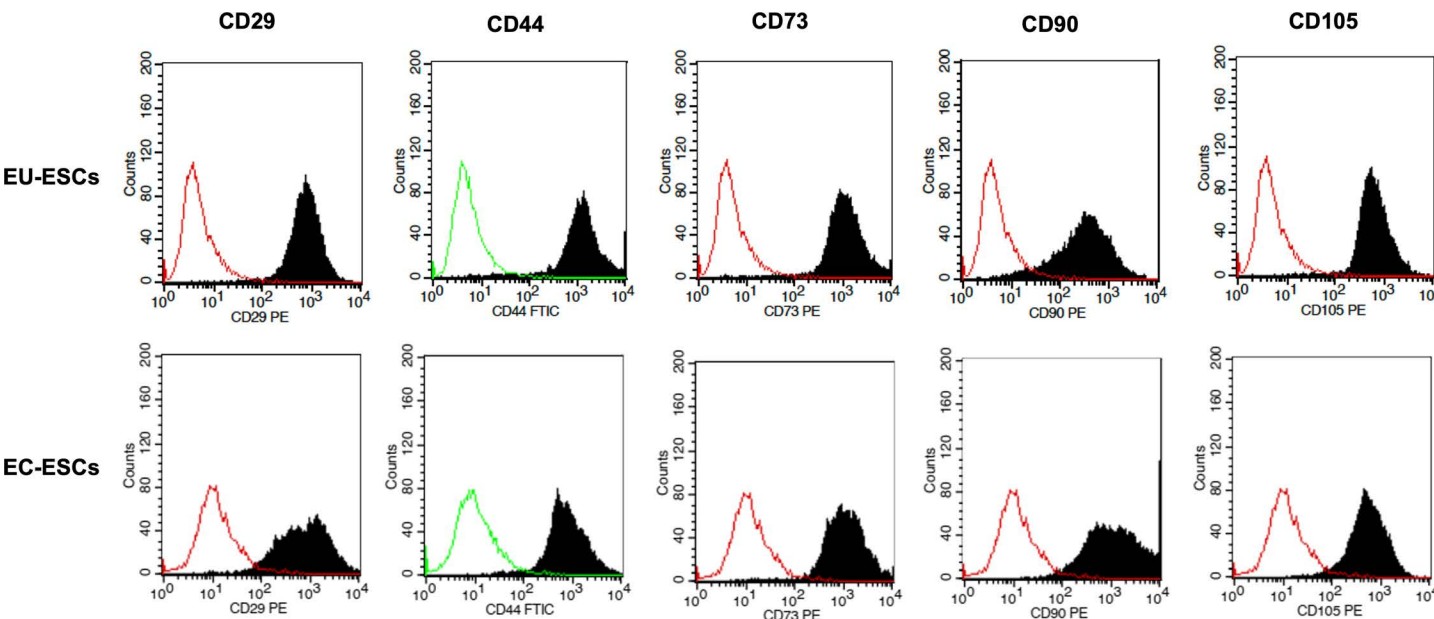

**Fig 3. Immunophenotypic of endometrial stromal cells.** Immunophenotyping of EU-ESCs and EC-ESCs was characterized by flow cytometry analysis. EU-ESCs and EC-ESCs were tested against mesenchymal stem cells markers: CD29, CD44, CD73, CD90, CD105. EU-ESCs and EC-ESCs were positive for the expression of typical MSC markers. EU-ESCs; Eutopic human endometrial stromal cells, EC-ESCs; Ectopic human endometrial stromal cells, CD; Cluster of Differentiation.

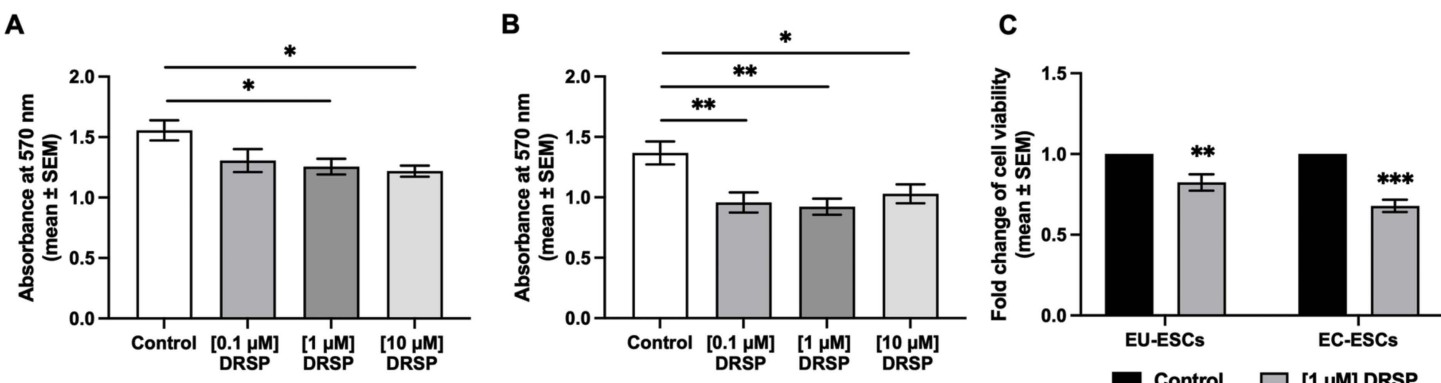

**Fig 4. Cell viability of EU-ESCs and EC-ESCs. (A)** MTT viability assay of EU-ESCs **(B)** MTT viability assay of EC-ESCs treated with 0.1, 1, and 10 µM of drospirenone. The mean±SEM (n=8) of absorbance was shown in a bar chart and analyzed with one-way ANOVA, with * $P < 0.05$ and ** $P < 0.01$ indicating statistical significance. **(C)** Cell viability fold change in EU-ESCs and EC-ESCs after treatment with 1 µM of drospirenone. Comparison of cell viability between the control group and treated groups were shown in a bar chart (mean±SEM, n=8) and analyzed with t-test, with ** $P < 0.01$ and *** $P < 0.001$ indicating statistical significance. DRSP; Drospirenone, EU-ESCs; Eutopic human endometrial stromal cells, EC-ESCs; Ectopic human endometrial stromal cells.

Drospirenone at 1 µM of concentration was considered to significantly effect cell viability on both EU-ESCs and EC-ESCs. The reduction in cell viability was obviously observed in 1 and 10 µM concentrations of drospirenone. At 1 µM concentration of drospirenone, the reduction of cell viability fold changes compared with the control untreated group were 0.810 (p=0.004) and 0.670 (p<0.001) in EU-ESCs and EC-ESCs, respectively (Fig 4C). Therefore, the lowest effective dose should be 1 µM that would be used for further analysis of Ki-67 and caspase-3 expression.

## Antiproliferation effect of drospirenone on EU-ESCs and EC-ESCs

The antiproliferation effect of drospirenone was evaluated using Ki-67 expression fold change of EU-ESCs and EC-ESCs treated with 1 µM concentration of drospirenone compared with the control group. In the EU-ESCs group, Ki-67 expression was reduced significantly after being treated with drospirenone (0.017 VS 1; p=0.003). On another hand, Ki-67 expression was slightly reduced in the EC group but without statistical significance (p=0.062) ([Fig 5]).

## Apoptotic Effect of Drospirenone on EU-ESCs and EC-ESCs

The apoptotic activity of EU-ESCs and EC-ESCs was determined by using caspase-3 expression with flow cytometry. After 24 hours of treatment with 1 µM of drospirenone, caspase-3 expression fold change was increased in both EU-ESCs and EC-ESCs compared with the control group with statistical significance (1.13 vs 1 and 1.57 vs 1, p=0.02 and p=0.05, respectively) ([Fig 6]).

In a quantitative RT-PCT study, the gene expression of apoptosis and tumor suppressor genes *BCL2, BAX, BAK, PTEN* and *P53* in EU-ESCs and EC-ESCs that were cultured under drospirenone were analyzed using qRT-PCR. After 24 hours of treatment with 1 µM of drospirenone, the expression of *BCL2* was lower compared to control medium. There was an increase in the expression of *BAX* in EU-ESCs and EC-ESCs treated with drospirenone compared to control medium. The expression of *PTEN* and *P53* were higher in EU-ESCs treated with drospirenone compared to control. On the other hand, EC-ESCs treated with drospirenone, the expression of *PTEN* and *P53* were significantly higher (p=0.03 and p=0.04, respectively) compared to untreated EC-ESCs. For *BAK* expression, after being treated with drospirenone, there was lower expression in EU-ESCs but higher expression in EC-ESCs compared to the control medium ([Fig 7]).

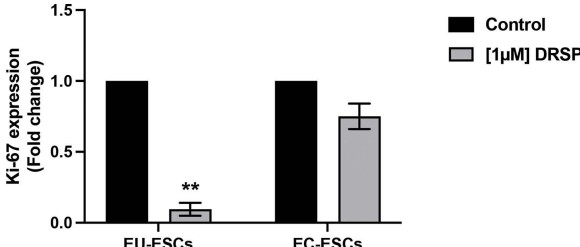

**Fig 5. Ki-67 expression fold change of EU-ESCs and EC-ESCs after being treated with 1 µM of drospirenone for 24 hours.** Fold change in both EU-ESCs and EC-ESCs compared to the control were shown in a bar chart (mean±SEM, n=3) and analyzed with t-test. Statistical significance was found when compared with the control group, ** *P*<0.01. DRSP; Drospirenone, EU-ESCs; Eutopic human endometrial stromal cells, EC-ESCs; Ectopic human endometrial stromal cells.

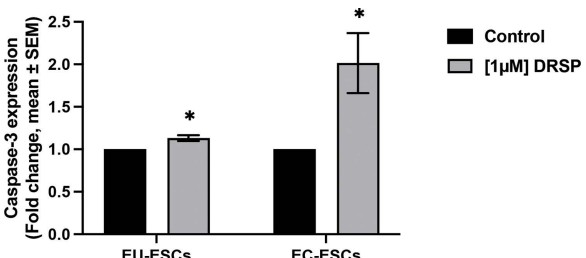

**Fig 6. Caspase-3 expression fold change of EU-ESCs and EC-ESCs after being treated with 1 µM of drospirenone for 24 hours.** Fold change in EU-ESCs and EC-ESCs compared to control were shown in a bar chart (mean±SEM, n=3) and analyzed with t-test. Statistical significance was found when compared with the control group. DRSP; Drospirenone, EU-ESCs; Eutopic human endometrial stromal cells, EC-ESCs; Ectopic human endometrial stromal cells. * *P*<0.05.

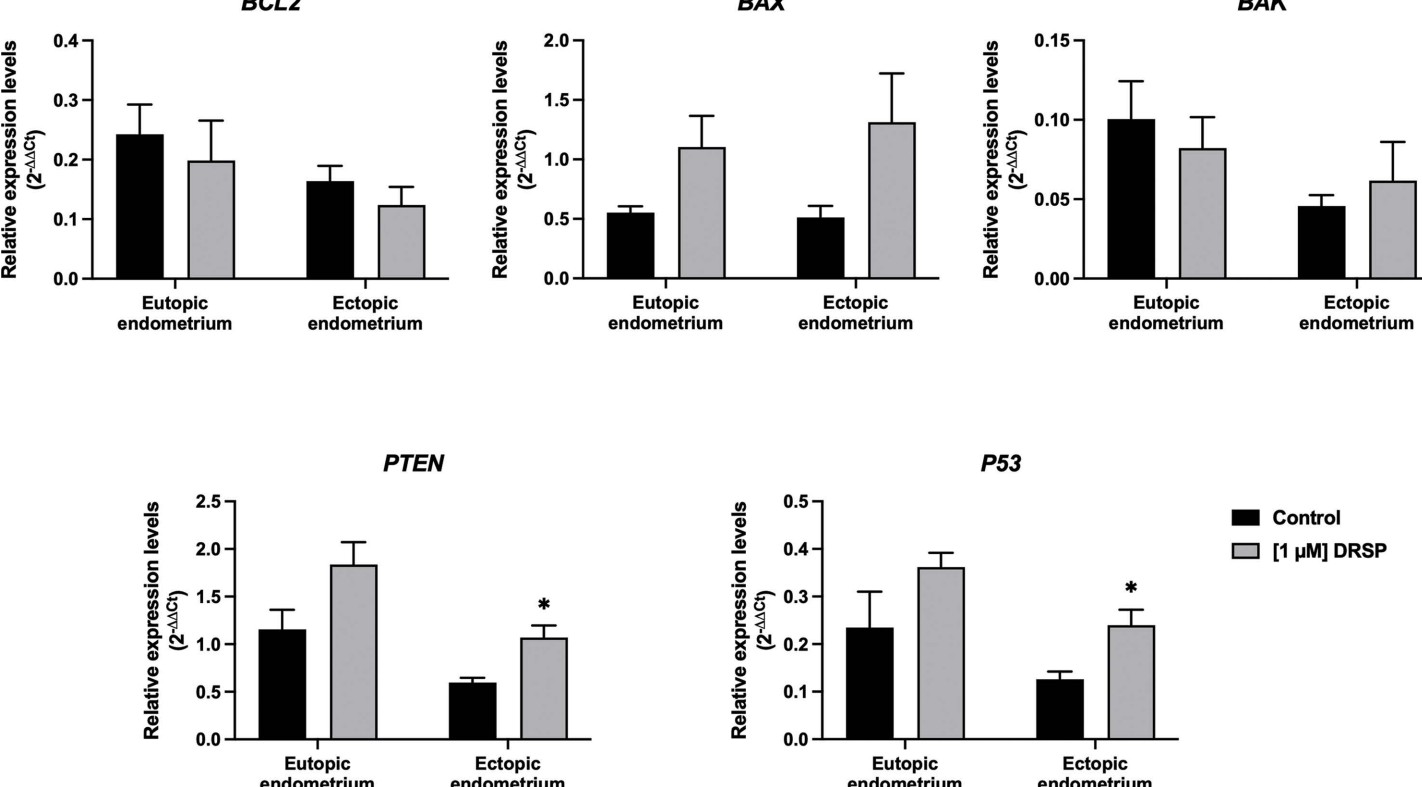

**Fig 7. The relative gene expression levels in EU-ESCs and EC-ESCs cultured for 24 hours under drospirenone were presented in the mean±SEM (n = 3) and analyzed with t-test.** Statistical significance was found when compared with the control group, * $P < 0.05$. DRSP; Drospirenone, EU-ESCs; Eutopic human endometrial stromal cells, EC-ESCs; Ectopic human endometrial stromal cells.

## Discussion

Progestins are a well-established treatment for endometriosis. Since endometriosis is a chronic condition requiring long-term management throughout a woman's reproductive life, various types of progestins have been developed to minimize unwanted effects, such as androgenic effects. Among these, drospirenone is a newer progestin introduced for the treatment of endometriosis. A continuous oral drosprinone regimen, which induces amenorrhoea by suppressing ovulation, has been shown to alleviate endometriosis-related symptoms such as dysmenorrhoea and dyspeunia, and to prevent the sequelae of disease progression. Furthermore, several in vitro studies have demonstrated the promising effect of progestins on both endometriosis and EU-ESCs. Notably, the chemical structure of drospirenone is derived from spironolactone, making it structurally distinct from other progestins. This study aims to investigate the in vitro effects of drospirenone on paired ectoic and EU-ESCs, focusing on its antiproliferation and pro-apoptosis properties.

In the study, the effect of drospirenone treatment on the viability of ESCs was assessed using MTT assay. Our findings showed that drospirenone reduced cell viability in a dose-dependent manner in both EC-ESCs and EU-ESCs. Notably, drospirenone exerted a more pronounced effect on EC-ESCs compared to EU-ESCs. Similar to other progestins, these findings confirm that drospirenone can reduce the viability of endometriotic cells in vitro, potentially leading to a reduction in endometriotic lesions in vivo. Interestingly, a drospirenone concentration of 1 µM corresponds to approximately 36.65 ng/mL Pharmacokinetic studies evaluating the therapeutic use of drospirenone in endometriosis have shown that a 3-mg drospirenone dose in a combined oral contraceptive pill achieves steady-state serum levels of approximately 20–25 ng/

mL, while a 4-mg drospirenone-only pill produces a peak serum concentration of about 41 ng/mL(20). Given that a serum drospirenone level of approximately 30 ng/mL is considered sufficient for endometriosis suppression, the concentration used in this *in vitro* experiment is reasonable and clinically relevant [20].

Using progestin in endometriosis treatment, antiproliferation and apoptosis of endometriotic human endometrial stromal cells have been confirmed in vitro by using dienogest, norethindrone [8,11]. However, the effect of drospirenone on EU-ESCs was demonstrated that the cells underwent decidualization, decreased proliferation, and a reduction of DNA synthesis [16]. That led to the question of whether the effect of drospirenone on EC-ESCs is similar to eutopic one. In the present study, the finding of anti-proliferation of drospirenone on EU-ESCs by a decrease in Ki-67 expression was similar to the previous study [16]. Interestingly, drospirenone was, in the first time, shown no significant effect on the anti-proliferation of EC-ESCs. These findings were different from other progestins [8,9,11].

In this study, the present study found that treatment with drospirenone can induced apoptosis in both EU-ESCs and EC-ESCs. But the apoptosis effect on EC-ESCs was more prominent than EU-ESCs. However, these findings contrast with a previous study, which reported that drospirenone-treated EC-ESCs did not show a significant increase in apoptosis compared with controls [16]. A potential explanation for this discrepancy lies in the different apoptotic markers used. While Miyashita *et al*. employed annexin V as their apoptotic marker, our study utilized caspase-3. Although both annexin V and caspase assays are widely used to detect apoptosis, they assess different stages of the apoptotic process. Annexin V binds to phosphatidylserine on the external surface of the plasma membrane—an early event in apoptosis—but this marker can also be present in necrotic cells. In contrast, caspase-3 activation occurs during the later execution phase of apoptosis and is considered a more specific indicator of apoptotic cell death [21–23]. This difference may explain our findings: drospirenone-treated cells may undergo apoptosis during exposure, but detection of late-stage apoptosis or progression to necrosis could require a longer duration. Moreover, our study also demonstrates

Moreover, we found that the expression levels of *P53* and *PTEN* were elevated in drospirenone-treated cells compared to non-treated controls; however, the differences reached statistical significance only in EC-ESCs. Both genes are well known for their roles in regulating programmed cell death [24], suggesting that drospirenone may effectively induce apoptosis in EC-ESCs and potentially contribute to the therapeutic control of endometriosis.

*PTEN* promotes apoptosis primarily by inhibiting the PI3K-Akt signaling pathway [24], while *P53* induces cell cycle arrest and activates apoptosis via both the intrinsic and extrinsic pathways [25]. The intrinsic pathway involves the upregulation of pro-apoptotic proteins such as *BCL2*, *BAX*, and *BAK*, leading to mitochondrial outer membrane permeabilization (MOMP) [25]. In contrast, the extrinsic pathway is initiated by activation of death receptors at the cell surface, which triggers downstream activation of effector caspases such as caspase-3 [26,27].

Interestingly, no significant differences were observed in the expression of *BCL2*, *BAX*, or *BAK* – key regulators of the intrinsic mitochondrial pathway -between drospirenone-treated and untreated EU-ESCs and EC-ESCs. In contrast, caspase-3 expression, associated with activation of the extrinsic apoptotic pathway, was significantly increased in the drospirenone-treated groups. Together with the observed changes in *P53* expression, these findings suggest that drospirenone predominantly induces apoptosis via the extrinsic pathway rather than through the intrinsic mitochondrial pathway.

This *in vitro* study is the first to demonstrate that drospirenone exerts an antiproliferative effect specifically on EU-ESCs, but not on EC-ESCs. Instead, drospirenone induces a more prominent apoptotic effect in EC-ESCs, likely mediated through upregulation of *PTEN* and *P53*, and activation of the extrinsic apoptotic pathway. These findings suggest that the biological effects of drospirenone differ from those of other progestins commonly used in the treatment of endometriosis.

From a clinical perspective, endometriosis presents with a wide spectrum of severity, ranging from progressive dysmenorrhea to the development of endometriotic cysts. Based on our findings, drospirenone may modulate endometriosis progression through its antiproliferative and pro-apoptotic effects. Indeed, drospirenone-only pills and combined oral contraceptives containing drospirenone have been shown to be effective and practical treatments for endometriosis-related symptoms. However, their role in the management of endometriotic cysts remains controversial. Drospirenone may be



more suitable for patients with endometriosis associated with small endometriotic cysts, as it may enhance apoptosis of endometriotic cells but may not sufficiently suppress cyst progression due to limited antiproliferative effects. In contrast, large endometriotic cysts are typically managed surgically, given the higher efficacy and lower recurrence rates associated with surgical treatment.

Further studies are warranted to determine whether the combination of estrogen and drospirenone exerts similar or synergistic effects on human eutopic endometrium and endometriotic lesions. In addition, the effects of drospirenone on endometrial stem cells—using primary cultures or cell lines—should be explored in other biological aspects, such as cell-cycle regulation and cellular migration using migration assays. Such investigations would help bridge the gap between biological mechanisms and clinical outcomes, thereby clarifying drospirenone's therapeutic role in the comprehensive management of endometriosis.

## Conclusion

This study demonstrates that drospirenone exerts distinct effects on EU-ESCs and EC-ESCs via different mechanisms. Specifically, drospirenone exhibits an antiproliferative effect on EU-ESCs and induces a more pronounced apoptotic response in EC-ESCs. These findings highlight the potential of drospirenone as an effective therapeutic agent in the management of endometriosis.

## Supporting information

**S1 File. Data Supplement.**
(XLSX)

## Acknowledgments

The authors thank the Gynecologic Endocrinology Unit staff for their invaluable advice and Dr Saowaluck Hunnanggul of the Clinical Epidemiology Unit for her support with the statistical analyses.

## Author contributions

**Conceptualization:** Thanyarat Wongwananuruk, Somsin Petyim.

**Data curation:** Kittima Tungprasertpol, Supatra Klaymook.

**Investigation:** Kittima Tungprasertpol, Supatra Klaymook.

**Methodology:** Thanyarat Wongwananuruk, Pawitra Suwannalert.

**Software:** Pawitra Suwannalert, Kittima Tungprasertpol, Supatra Klaymook.

**Supervision:** Somsin Petyim.

**Writing – original draft:** Thanyarat Wongwananuruk, Pawitra Suwannalert.

**Writing – review & editing:** Somsin Petyim.

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
