## [Decision Letter · Decision Letter 0]

17 Nov 2025

Dear Dr. Petyim,

Thank you for submitting your manuscript to PLOS ONE. After careful consideration, we feel that it has merit but does not fully meet PLOS ONE’s publication criteria as it currently stands. Therefore, we invite you to submit a revised version of the manuscript that addresses the points raised during the review process.

We look forward to receiving your revised manuscript.

Kind regards,

Stefania Crispi

Academic Editor

PLOS ONE

2. In the online submission form, you indicated that all the data analyzed during this study are available from the corresponding author upon reasonable request.

This study was supported by the Research Development Grants, Faculty of

Medicine Siriraj Hospital, Mahidol University (no. R016631035).

Reviewers' comments:

Reviewer's Responses to Questions

**Comments to the Author**

1. Is the manuscript technically sound, and do the data support the conclusions?

Reviewer #1: Yes

Reviewer #2: Partly

2. Has the statistical analysis been performed appropriately and rigorously?

Reviewer #1: Yes

Reviewer #2: Yes

3. Have the authors made all data underlying the findings in their manuscript fully available?

Reviewer #1: Yes

Reviewer #2: Yes

4. Is the manuscript presented in an intelligible fashion and written in standard English?

Reviewer #1: Yes

Reviewer #2: Yes

Reviewer #1: The authors with improved methodological transparency, statistical justification, and language editing, the manuscript could make a strong contribution to the field. The biological insights are interesting and potentially impactful, but the presentation and interpretation need refinement for clarity.

The introduction effectively summarizes the background but repeats information (e.g., progestin effects on apoptosis). Please streamline to highlight the knowledge gap specifically, why drospirenone’s distinct chemical structure warrants this study.

The “Materials and Methods” section provides substantial procedural detail, but lacks clarity on certain aspects:

- Specify the number of biological replicates and technical replicates for each assay.

- Clarify the rationale for selecting 1 µM drospirenone—mention how this concentration relates to physiological or pharmacological levels.

- Indicate how cell purity was verified (e.g., immunostaining for stromal markers such as vimentin or absence of epithelial markers).

- Has a characterization of the cells isolated from the patients been done?

While statistical methods are mentioned, there is no justification for using parametric tests with such small sample sizes (n=8 effective samples). Consider verifying data normality or using non-parametric tests. Include sample size and variance information in figure legends.

Figures and results are clearly structured, but key data need better quantification:

- Report exact p-values instead of only “p < 0.05”.

- Include mean ± SEM or box plots for all bar charts.

- Describe Figure 6 results more precisely—state whether fold changes were normalized to controls.

The discussion is comprehensive but overly descriptive and partially redundant.

- Clearly differentiate between your findings and those from previous studies.

- The explanation of apoptotic pathways (intrinsic vs extrinsic) is useful but needs more direct connection to your gene-expression data.

- Include a short paragraph on clinical relevance (e.g., how these in vitro results may inform treatment regimens).

The manuscript is generally understandable but requires language polishing to meet journal standards.

- Remove unnecessary periods after abbreviations (e.g., “.In many in vitro studies” → “In many in vitro studies”).

- Correct spacing and grammar inconsistencies (“10%.DMEM” → “10% DMEM”).

- Consider professional English editing for smoother readability.

Minor Comments

1. Line 70–74: “Endometriosis is a complex gynecological condition...” → Combine short sentences for better flow.

2. Line 105–120: Clarify inclusion/exclusion criteria and ethical consent procedures.

3. Table 1: The forward primer for BCL2 is missing; check for duplication (“R:” repeated twice).

4. Figures: Ensure consistent file naming (“Figure 6” not “Figure” only).

5. Add abbreviation list after the abstract or before references (e.g., EU-ESCs, EC-ESCs, DRSP).

6. Reference section: Check formatting according to PLOS ONE guidelines journal names should be italicized and consistent.

Reviewer #2: The article is interetsing and presents original data. I have some concerns that should be addressed by the authors; the manuscript should be modified and implemented accordingly

1) The authors should perform some preliminary studies on the isolated cells in order to better define their phenotype, such as analyze the expression of estrogen and progesteron, and expression of stromal markers such as CD10. Moreover, to investigate the effect of immortalization on cell karyotypes, cytogenetic analysis should be performed using standard

G-banding analysis

2) In order to better define the effect of Drospirenone, I suggest to define effects on cell cycle by FACS and the effects on celle mobility by a migration assay

**Do you want your identity to be public for this peer review?** For information about this choice, including consent withdrawal, please see our Privacy Policy

Reviewer #1: No

Reviewer #2: No

---

## [Author Response · Author response to Decision Letter 1]

23 Dec 2025

Response to reviewers

Review Comments to the Author

Comments to the Author

Reviewer #1:

1.1. The authors with improved methodological transparency, statistical justification, and language editing, the manuscript could make a strong contribution to the field. The biological insights are interesting and potentially impactful, but the presentation and interpretation need refinement for clarity. The introduction effectively summarizes the background but repeats information (e.g., progestin effects on apoptosis). Please streamline to highlight the knowledge gap specifically, why drospirenone’s distinct chemical structure warrants this study.

Answer: Thank you for your comments. I agreed with your comment, and we have changed and streamlined to highlight the knowledge gap specifically. (mentioned in lines 95-112, 120-127)

1.2. The “Materials and Methods” section provides substantial procedural detail, but lacks clarity on certain aspects:

- Specify the number of biological replicates and technical replicates for each assay.

Answer: Thank you for your comments. We have checked and corrected as your suggestion. (mentioned in Lines 148-149)

- Clarify the rationale for selecting 1 µM drospirenone—mention how this concentration relates to physiological or pharmacological levels.

Answer: Thank you for your comments. We add the rationale for selecting 1 µM drospirenone- relates to physiological or pharmacological levels as your suggestion. (mentioned in Lines 371-378)

- Indicate how cell purity was verified (e.g., immunostaining for stromal markers such as vimentin or absence of epithelial markers). Has a characterization of the cells isolated from the patients been done?

Answer: Thank you for your comments. Your comment is very useful. In this study, endometrial stromal cells were isolated and phenotypically characterized as endometrial stromal stem cells, and their surface markers were identified. This data has been added in Figure 3. (mentioned in Lines 172-174, 267-268)

- While statistical methods are mentioned, there is no justification for using parametric tests with such small sample sizes (n=8 effective samples). Consider verifying data normality or using non-parametric tests. Include sample size and variance information in figure legends.

Answer: Thank you for your comments. I agree with your comment. In the study, all statistical calculations were performed by a statistician. We added the information in the part of Statistical Analysis, and added sample size and variance information in the figure legends as your suggestions. (mentioned in lines 246-248, 299-307, 315-320, 328-333,347-351)

1.3. Figures and results are clearly structured, but key data need better quantification:

- Report exact p-values instead of only “p < 0.05”.

Answer: Thank you for your comments. We have checked and corrected as your suggestions. (mentioned in Lines 62, 64, 67-68, 271, 273-275, 280.313. 314. 325, 342)

- Include mean ± SEM or box plots for all bar charts.

Answer: Thank you for your comments. We have checked and corrected as your suggestions. (mentioned in Lines 300-301, 304, 317, 330, 348)

- Describe Figure 6 results more precisely—state whether fold changes were normalized to controls.

Answer: Thank you for your comments. We have checked figures 5 and 6, corrected as your suggestion. (mentioned in Lines 315-320, 328-333)

1.4. The discussion is comprehensive but overly descriptive and partially redundant.

- Clearly differentiate between your findings and those from previous studies.

Answer: Thank you for your comments. We have corrected as your suggestion. (mentioned in Lines 388-399)

- The explanation of apoptotic pathways (intrinsic vs extrinsic) is useful but needs more direct connection to your gene-expression data.

Answer: Thank you for your comments. We have corrected as your suggestion. (mentioned in Lines 414-420)

- Include a short paragraph on clinical relevance (e.g., how these in vitro results may inform treatment regimens).

Answer: Thank you for your comments. We added a short paragraph on clinical relevance as your suggestion. (mentioned in Lines 427-437)

The manuscript is generally understandable but requires language polishing to meet journal standards.

- Remove unnecessary periods after abbreviations (e.g., “.In many in vitro studies” → “In many in vitro studies”).

Answer: Thank you for your comments. We have checked and corrected as your suggestion. (mentioned in Lineห 49, 110)

- Correct spacing and grammar inconsistencies (“10%.DMEM” → “10% DMEM”).

Answer: Thank you for your comments. We have checked and corrected as your suggestion. (mentioned in Line 162)

- Consider professional English editing for smoother readability.

Answer: Thank you for your comments. I have enclosed the letter of English editing certification. The title of the study in the letter was the previous version (Title: Effect of Drospirenone on Anti-proliferation and Apoptosis in Ectopic Endometrial Stromal Cells: an In Vitro Study”).

1.5. Minor Comments

1. Line 70–74: “Endometriosis is a complex gynecological condition...” → Combine short sentences for better flow.

Answer: Thank you for your comments. We have changed as your suggestions. (mentioned in Lines 78-79)

2. Line 105–120: Clarify inclusion/exclusion criteria and ethical consent procedures.

Answer: Thank you for your comments. We have changed as your suggestions. (mentioned in Lines 130-140)

3. Table 1: The forward primer for BCL2 is missing; check for duplication (“R:” repeated twice).

Answer: Thank you for your comments. It was corrected as mentioned in Line 243.

4. Figures: Ensure consistent file naming (“Figure 6” not “Figure” only).

Answer: Thank you for your comments. Figure 6 is changed to Figure 7. We have checked and corrected as your suggestion in Line 345.

5. Add abbreviation list after the abstract or before references (e.g., EU-ESCs, EC-ESCs, DRSP).

Answer: Thank you for your comments. We added the abbreviation list as your suggestion. (mentioned in Lines 458-460)

6. Reference section: Check formatting according to PLOS ONE guidelines journal names should be italicized and consistent.

Answer: Thank you for your comments. The reference was done in Vancouver style as the author guidelines.

Reviewer #2: The article is interetsing and presents original data. I have some concerns that should be addressed by the authors; the manuscript should be modified and implemented accordingly

2.1. The authors should perform some preliminary studies on the isolated cells in order to better define their phenotype, such as analyze the expression of estrogen and progesteron, and expression of stromal markers such as CD10. Moreover, to investigate the effect of immortalization on cell karyotypes, cytogenetic analysis should be performed using standard G-banding analysis

Answer: Thank you for your comments. In this study, endometrial stromal cells were isolated and phenotypically characterized as endometrial stromal stem cells, and their surface markers were identified. This data has been added in Figure 3. Moreover, the endometrial stromal cells used were primary cells rather than immortalized cells or established cell lines; therefore, karyotype or cytogenetic analyses were not performed. This has been added to the future study. (mentioned in Lines 267-268, 293-297, 440-442)

2.2. In order to better define the effect of Drospirenone, I suggest to define effects on cell cycle by FACS and the effects on cell mobility by a migration assay

Answer: Thank you for your comments. Your suggestions are very useful. We did not perform cell cycle or cell mobility analyses in this study, and this has been added to the discussion part and further study. (mentioned in Lines 440-442)

---

## [Decision Letter · Decision Letter 1]

8 Jan 2026

Drospirenone promotes apoptosis in ectopic but inhibits proliferation in eutopic human endometrial stromal cells

PONE-D-25-51558R1

Dear Dr. Somsin Petyim,

We’re pleased to inform you that your manuscript has been judged scientifically suitable for publication and will be formally accepted for publication once it meets all outstanding technical requirements.

Kind regards,

Stefania Crispi

Academic Editor

PLOS One

Additional Editor Comments (optional):

All the reviewers comments have been addressed

Reviewers' comments:

Reviewer's Responses to Questions

**Comments to the Author**

Reviewer #1: (No Response)

Reviewer #2: All comments have been addressed

2. Is the manuscript technically sound, and do the data support the conclusions?

Reviewer #1: (No Response)

Reviewer #2: Yes

3. Has the statistical analysis been performed appropriately and rigorously?

Reviewer #1: (No Response)

Reviewer #2: I Don't Know

4. Have the authors made all data underlying the findings in their manuscript fully available?

Reviewer #1: (No Response)

Reviewer #2: Yes

5. Is the manuscript presented in an intelligible fashion and written in standard English?

Reviewer #1: (No Response)

Reviewer #2: Yes

Reviewer #1: (No Response)

Reviewer #2: (No Response)

**Do you want your identity to be public for this peer review?** For information about this choice, including consent withdrawal, please see our Privacy Policy

Reviewer #1: No

Reviewer #2: **Yes:** Alfonso Baldi

---

## [Editor Report · Acceptance letter]

PONE-D-25-51558R1

PLOS One

Dear Dr. Petyim,

I'm pleased to inform you that your manuscript has been deemed suitable for publication in PLOS One. Congratulations! Your manuscript is now being handed over to our production team.

Kind regards,

on behalf of

Dr. Stefania Crispi

Academic Editor

PLOS One